



# The influence of climate variability on transatlantic flight times

Corwin J. Wright[1], Phoebe E. Noble[1], Timothy P. Banyard[2,1], Sarah J. Freeman[3], and Paul D. Williams[4]

[1]Centre for Climate Adaptation and Environment Research, University of Bath, Bath, UK
[2]Department of Earth and Environmental Sciences, University of Manchester, Manchester, UK
[3]Nature Positive Ltd, Bath, UK
[4]Department of Meteorology, University of Reading, Reading, UK

**Correspondence:** Corwin Wright (c.wright@bath.ac.uk)

**Abstract.** Transatlantic aviation is a major industry and even small flight time changes have major economic and environmental implications. While our ability to optimise these flights for background wind variations at day-to-day scales is excellent, at the longer timescales needed for sustainability planning and fuel cost hedging these capabilities are more limited. Here, we quantify the association between four climate indices (the El Niño-Southern Oscillation, the North Atlantic Oscillation, the Quasi-Biennial Oscillation and solar irradiance) and transatlantic flight times using thirty years of commercial flight data. This allows us to identify whether these indices can be used to identify systematic flight time shifts. We find that ENSO and the NAO are associated with statistically-significant changes in one-way flight times of up to 82.2±3.5 minutes, and changes in round-trip times of 4.8±0.5 minutes and 4.0±0.8 minutes respectively, while the QBO and TSI have weaker but significant effects. Together, these indices plus a linear trend explain up to 27% of variation depending on season and direction, and are associated with month-to-month fuel cost & CO2 emission variations of up to 27MUSD & 120kT for one-way trips and 5 million USD & 23kT for round trips. We also show that westward, round-trip and non-winter-eastward flight times have increased by several minutes per decade since the 1990s, and that flights fly two-thirds of a standard deviation higher in altitude during solar maximum. Our results provide the first observational quantitative basis for aviation fuel and carbon cost management at monthly and longer timescales.

## 1 Introduction

Commercial flights in the transatlantic corridor between Europe and North America are a major component in the global trade and travel network, with hundreds of individual flights transiting the region every day. Due to the large number of these flights and the significant expense of operating them in both financial and climate terms, even small changes in transit times have very significant economic and environmental implications (e.g. Kim et al., 2020; Lee et al., 2021; Wells et al., 2021, 2022, 2023).

The winds through which a plane flies are a major factor in flight times. In the Atlantic sector, the dominant wind pattern is the northern midlatitude jet stream (hereafter the 'jet'), and accordingly modern flight planning uses forecasts of jet location and speed to optimise flight times, particularly in winter when the jet is strongest. Routes are optimised on the basis of assimilative numerical weather prediction (NWP) forecasts made at the start of a flying day, and due to the high quality of modern NWP



systems these flights now typically operate with only a few percent of deviation from the perfect route (Wells et al., 2021;
Boucher et al., 2023).

At longer timescales, however, the accurate prediction of flight-level winds remains highly challenging, with major uncertainties and biases remaining in most models of the upper troposphere and lower stratosphere (UTLS) at lead times longer than a few days (e.g. Lawrence et al., 2022). Such timescales are not important for day-to-day route optimisation, but predicting systematic flight time changes at these long timescales is important for longer-term operational choices and strategies such as
fuel price hedging, choosing whether to deploy a specific plane to one route versus another, and broader corporate sustainability planning.

Since numerical weather predictions of wind speed and jet morphology at these timescales are challenging to produce, an alternative approach is to study climate processes that have been shown to teleconnect with and drive the jet and its surrounding winds, such as the El Niño-Southern Oscillation (ENSO) or the stratospheric quasi-biennial oscillation (QBO) (e.g. Hall et al.,
2014; Domeisen et al., 2019; Anstey et al., 2021b; Kumar et al., 2022; Alizadeh, 2023). While their effects on the jet are weaker and more uncertain than those of day-to-day weather variations, the time taken for them to influence the jet means that they can provide useful information on statistically-likely shifts in wind speeds, and hence travel times, with a large lead time. Similarly, the North Atlantic Oscillation (NAO), a pressure pattern index which is linked to the jet, has been shown to be predictable at scales of a month or more (Strommen, 2020; Collingwood et al., 2024) and thus can also act as a source of predictability, while
solar irradiance has been suggested by several studies to play a role in Atlantic winds (e.g. Hall et al., 2014; Gray et al., 2013) and is predictable at the decadal timescale.

Accordingly, we here quantify the association between real measured transatlantic flight times obtained from aircraft data and four climate system indices, i.e. (a) ENSO, (b) the NAO, (c) the QBO, and (d) the 11-year solar output cycle, together with (e) a linear trend and (f) a sinusoidal annual cycle. Our aircraft data are obtained from the In-Service Aircraft for a Global
Observing System (IAGOS) program (Boulanger et al., 2019), which has operated measurement equipment on commercial aircraft since 1994. After quality control and regional subsetting our dataset contains 16 327 flights spread across 10 835 days between August 1994 and March 2024, providing a long time series and allowing us to study these the role of these four indices over multiple full cycles.

Section 2 describes the flight track and climate index data we use and Section 3 our selection, standardisation and quality
control procedures, including how we estimate round-trip times based on pairs of closely-spaced flights. In Section 4 we then assess the data at a broad level, including estimating 30-year linear trends and the bulk differences between climate index extrema. In Section 5 we apply multilinear regression techniques to quantify relative travel times at the individual flight level, and then use the results of this to estimate the cost impacts of these changed to travel time in terms of additional emissions of carbon dioxide and financial costs in Section 6. Finally, in Section 7 we use the same dataset to quantify links between flight
altitude and climate processes, before contextualising our results and drawing conclusions in Sections 8 and 9.





## 2 Data

### 2.1 IAGOS

We use data from IAGOS (Boulanger et al., 2019), which fits instruments to commercial aircraft to measure UTLS chemistry. In this study, we use only their metadata, specifically the times, latitudes, and longitudes recorded by the aircraft during each flight. We analyse all data available as of the 1st of September 2024 in the period from the 1st of August 1994 to the 30th of March 2024. The full dataset contains 67 289 flights, of which we analyse a subset of 16 327 individual flights selected as described in Section 3.1. Figure 1a shows the number of flights per month, made by a total of thirteen unique aircraft. We also show 'round trips', which we define in Section 3.3 below.

There are some key advantages to using this dataset. These include (1) that the data are fully open-access[1] and hence our approach can be easily reproduced and modified for the study of other regions and processes, (2) that a multi-decadal time series of flight data is available and is expected to extend into the future, (3) that the data are traceable to specific aircraft allowing for robust consistency checking and quantitive assessment of their fuel use, and (4) that we have access to colocated measurements of wind and chemistry from the aircraft to support future investigations into related processes.

There are however some disadvantages, the most significant of which is that we are unable to control for operational choices affecting flight time, for example aircraft speeding up or slowing down to meet a schedule. For the purposes of this study we assume that such effects cancel over a sufficiently long time series and varied set of routes and conditions, but this is unlikely to be completely true.

In terms of flight optimality, Boucher et al. (2023) used recorded IAGOS tracks and ERA5 reanalysis winds for the period 2018-19 to demonstrate that flight tracks in the IAGOS dataset were on average only around 1% longer than a computational optimum for transatlantic routes. To produce this estimate, while they used the actual trajectories flown, they did not use the recorded flight times but instead re-estimated these under assumptions of constant airspeed (240 ms$^{-1}$) at the mean cruising altitude level of the specific flight, thus removing the impact of operational choices.

### 2.2 Climate Indices

We compare these flight metadata to six climate-system indices, in order to quantify any relationship between the processes these indices characterise and IAGOS flight times. The indices are all defined as one-dimensional time series with a single value for each day. They are:

- 'ENSO', the Nino3.4 index (Trenberth and Stepaniak, 2001) characterising the El Niño-Southern Oscillation. ENSO is well-known to have global effects on wind patterns, and has been shown to be associated with gravity wave variability over the North Atlantic in IAGOS data (Wright and Banyard, 2020), and to impact upon simulated trans-Pacific flight times estimated using reanalysis wind fields (Kim et al., 2020). Large values are defined as El Niño and small values as La Niña.

---

[1]Specifically, the data we use are licensed as CC-BY 4.0 at time of writing, i.e. free for sharing and adaptation subject to attribution.



**Figure 1.** (top panel) Number of flights per month in the dataset. Each row represents an individual plane, identified by tail number at far left, with bubble sizes showing total flights by that plane each month, summed at right. The top dataset, labelled 'round', is the number of round trips each month as defined in Section 3.3 (remaining panels) Time series of normalised indices used in our analyses. Date ticks indicate first day of year, and the original full range of the data is indicated at the right-hand side of each panel, with units indicated at right-centre of each panel where appropriate.

- 'NAO', the North Atlantic Oscillation (Hurrell et al., 2003). The NAO is the dominant mode of surface-level pressure variability in the North Atlantic, and a positive NAO is associated with poleward shifting of the polar front jet. The NAO is thus expected to directly and strongly affect flight times. It was also shown to impact reanalysis-inferred transatlantic flight times in the reanalysis-based study of Kim et al. (2020), who discuss in detail the physical mechanisms by which such interactions occur. Large positive values are defined as NAO+ and large negative values as NAO-.

- 'QBO', representing the equatorial Quasi Biennial Oscillation. The QBO is the dominant source of interannual variability in the tropical stratosphere, and acts on North Atlantic jet speeds by confining planetary wave activity to the tropics. We





use a time series of monthly-mean zonal-mean zonal wind speeds within $\pm 5°$ of the Equator at the 50 hPa level derived

from ERA5 reanalysis output (Hersbach et al., 2020). Since we lag our indices (discussed below) and the QBO descends in altitude in a regular temporal pattern, the choice of the 50 hPa level should not strongly affect our results, as a change in level would simply lead to a corresponding change in our optimal-lag calculation. Large positive values are defined as QBO-westerly and large negative values as QBO-easterly.

– 'TSI', representing the total irradiance of the Sun as received by the Earth (Coddington et al., 2015). Previous work

suggests that solar-driven changes in upper-stratospheric ozone can propagate downwards and modulate the stratospheric polar night jet (Kodera and Kuroda, 2002), albeit at significant lead times (e.g. Gray et al., 2013; Scaife et al., 2013), and thus may play a role in flight times. Note that, as the input TSI data are extremely noisy at the daily level and we expect any influence to be slow, we have removed a single large negative outlier in 2003 and smoothed the remaining data using a 15-day boxcar filter. Large values are defined as solar maximum and small values as solar minimum.

– 'Time', a linear trend through the period studied. In Section 4.1 we demonstrate that the dataset exhibits significant linear trends. Consequently, we include this as an index in our other analyses to regress out linear-trend effects. This index may include some signal due to climate change, which we do not investigate separately.

– 'Annual', a sinusoid with a maximum of 1 on January 1st and a minimum of -1 on July 1st. This acts as a simple estimate of the seasonal cycle, and is only used in analyses where we do not explicitly subdivide the data into seasons.

For greater depth on the above, we refer the reader to Hall et al. (2014), who discuss in detail the physical processes these indices characterise and how they act on jet speeds in the North Atlantic corridor.

Figures 1b-g show these indices over the period 1994–2024. Each has been normalised to the range -1 to +1, where -1 is the lowest value reached and +1 the largest; the original range is shown at the right of each panel. All indices except Time exhibit multiple complete cycles and, additionally, all indices except TSI vary multiple times within the lifetime of individual aircraft

contributing to the dataset.

We do not expect the physical processes associated with these indices to act instantaneously on winds and hence flight times, and accordingly we lag the indices. To identify the most appropriate lag for each index, Appendix A1 describes a cross-correlation analysis carried out for lags of up to one year. Based on these results, we lag as follows:

– NAO: 0 days for westward-bound and eastward-bound flights, and 1 day for round-trip flights

– ENSO: 11 days westward, 0 days eastward, 35 days round-trips

– QBO: 187 days westward, 270 days eastward, 190 days round-trips

– TSI: 362 days westward, 258 days eastward, 360 days round-trips

Lagging the coefficients in this way is a methodological choice, and in supplementary material we include an analysis without any lag applied, discussed in Section 5 below. In Appendix A2 we quantify index independence statistically, concluding that the





combination we use is sufficiently free of multicollinearity and autocorrelation. We also considered the following indices, but do not use them in our study: (i) deseasonalised sea ice cover and sea surface temperature, both of which exhibit multicollinearity with other variables, (ii) wind speeds at 10 hPa and 60°N, which act as a metric of polar vortex strength in winter but otherwise often closely trace the annual cycle, and (iii) the hypothesised Atlantic Multidecadal Oscillation, discussed in Appendix B.

## 3  Methods

### 3.1  Data Selection and Subsetting

IAGOS records data globally (see e.g. Figure 2 of Wright and Banyard, 2020). Accordingly, to investigate the North Atlantic sector only, we must subset the flights.

We begin by identifying airports which are either the origin or destination of any IAGOS flight. We hand-classify each airport as being in western Europe (EUR), eastern North America (NA), or elsewhere, discarding the third set. We then further
filter by identifying all records originating in either the NA or EUR airport list and travelling to the other, again discarding all other flights, i.e. those internal to NA or Eur.

For the remaining records, we discard data within 10 km of the origin and destination to remove delays due to takeoff and landing; the sensitivity test used to select this value is discussed in Appendix C1. We then quality-control the data by sequentially applying filters to remove flights with discontinuities >15 minutes (408 flights), >10° of latitude (1 flight) and
>10° of longitude (96 flights) respectively.

We next split the data into individual 'routes', defined as flights from a specific departure airport to a specific arrival airport in one direction. Several routes have only a small number of flights, and to prevent these from skewing our results we remove any routes with <10 flights; this choice is sensitivity-tested in Appendix C2.

Figure 2a-b shows the 27 airports which remain after these filters, identified by IATA codes. For full context, Supplementary
Table 1 shows the number of flights between each airport-pair; the most common routes each way are between ATL and FRA, with 1139 ATL-to-FRA flights and 1132 FRA-to-ATL flights. We have separately tested all analyses using only these two routes, and our results remain broadly consistent but with larger statistical uncertainties, i.e. the choice to composite all routes does not significantly affect our results.

Figures 2c-l show the distribution of paths taken, for both the complete dataset and the four seasons DJF (boreal winter),
MAM (boreal spring), JJA (boreal summer) and SON (boreal autumn). The data have been subdivided into eastward and westward flights, then binned onto a one-second scale and summed onto a 0.2° grid to give a total flight time spent in each box[2] For context Figures 2m-p shows seasonal-mean 250 hPa ERA5 zonal winds over the period 1991-2020, with median flight traces in each direction overlaid.

Local maxima can be seen near individual airports such as FRA, ATL and JFK, consistent with the large fraction of flights
using them. Away from airports, in general westward flights traverse a wide range of latitudes whilst eastward flights have a

---

[2]Note that the area of each box varies as as the cosine of latitude, and hence maps of (e.g.) flights per square kilometre would show systematic differences from Figure 2c-l.





**Figure 2.** (a-b) Maps of arrival and departure airports meeting our selection criteria in (a) Eastern North America and (b) Western Europe. Three-letter identifying codes are shown geographically centred at the location of the airport; note that 'EWR' and 'JFK' (41°N, 71°W) overlap closely. (c-l) Maps of paths taken by flights in our analysed subset of the data, in the (c-g) westward and (h-l) eastward direction, for (c,h) the full dataset and seasons (d,i) DJF, (e,j) MAM, (f,k) JJA and (g,l) SON. Values are shown as the number of flight-minutes recorded in each 0.2° box by all flights in that direction and season. (m-p) maps of climatological (ERA5 averaged over the period 1991-2020) zonal winds at 250 hPa for each season. In panels (c-p), overlaid blue and black lines indicate the median latitude of all flights in the dataset at each longitude between 75°W and 5°W, shown in black for westward flights and blue for eastward.

narrower meridional distribution, and the median path in every season is consistently further south and nearer the jet centre for eastward flights. We see a fine mesh of overlapping lines, representing the North Atlantic Organised Track System (NAT-OTS). The data are consistent with aircraft taking advantage of high wind jet speeds in the eastward (i.e. downstream) direction





and avoiding the jet centre in the westward (i.e. upstream) direction, in turn driven by management of NAT-OTS using daily

numerical weather forecasts.

## 3.2   Travel Time Standardisation

The above preprocessing provides the actual time taken by each flight. However, such data is hard to compare between different airport-pairs due to the different distances involved. We have considered two possible solutions to this problem which allow standardisation while still retaining the broad perspective on transatlantic aviation available from considering multiple airport-

pairs.

One solution would be to remove all sections of a flight outside of the central component, e.g. from 60°W–20°W. This would remove travel time to inland airports, but not address the problem of comparability. For example, flights from Europe to ORD (O'Hare International Airport, 42°N, 88°W) flying great-circle routes closer to the pole will spent a smaller proportion of their flight time within this range than flights to ATL (34°N, 84°W).

Instead, we normalise flight times within each route, as defined above. To do so, we compute the median travel time for each route, then normalise each flight to this. To avoid outliers, we remove any flight with a travel time $<85\%$ or $>115\%$ of the route-median; in practice this only removes 1 flight. To aid interpretation, we then scale our results to the median flight over all 16 327 flights, which is 509.92 minutes. To do so, we multiply each normalised flight time by this median, then subtract the median value again to give a time deviation. We refer to this normalised deviation as the 'delay' of the flight relative to the

overall median.

As an example of the 'delay' calculation, consider a flight from AAA to BBB which takes 550 minutes against a median travel time from AAA to BBB of 500 minutes. We first normalise the travel time to produce a relative time of 1.10, then scale by the all-dataset median of 509.92 minutes to produce a notional delay of 50.92 minutes, i.e. 10% of the all-flights median time. This delay of 50.92 minutes is the value used in our subsequent analyses, rather than the true 50-minute difference in

flight time.

A possible side-effect of this approach is that the wide range of latitudes experienced could confound our results due to higher-latitude flights spending a larger fraction of flight time in the stratosphere. To test this, we have performed correlation analyses between flight delays and (a) the maximum latitude reached by each flight and (b) the proportion of each flight spent in the stratosphere, the latter using tropopause heights computed from ERA5 data following the method of Reichler et al. (2003).

For latitudes, Pearson linear correlations with delay range between 0.06 and 0.08, while for stratopause fractions they range between 0.07 and 0.08. This relatively small correlations suggests that this issue is relatively minor as a factor in our analysis.

## 3.3   Round-Trip Time

In some Sections we discuss 'round-trip' delays, i.e. net delays across a trip travelling across the Atlantic and back again. Such round-trip-residual delays can arise due to differences in routing or wind speeds in each direction, but still arise even in the case

of an identical route through steady winds as described by Williams (2016). Characterising these round-trip delays is useful





for identifying fuel use and $CO_2$ impacts of climate-process-delayed flights, since over a sufficient time-average aircraft will tend to return to their home base.

To estimate these, we generate a set of synthetic round trips from one-way IAGOS records using the following procedure:

1. We first identify flights with a North American airport of arrival. This choice is largely arbitrary, but is consistent with most aircraft in our analysis being European-registered

2. For each such flight, we identify the European origin airport.

3. We then find the temporally-closest flight in our record flying back from the North American airport to the same European origin and treat this pair of flights as a round-trip, including computing a net delay across the two legs.

We impose a one-day limit between arrival and departure at the North American airport; in practice the vast majority are separated by less than half a day. To maximise the number of such round trips, we (a) permit negative time differences, i.e. a 'round-trip' could 'return' before it 'arrives', and (b) allow the same flight to contribute to multiple synthetic round trips.

## 4 High-Level Analyses

### 4.1 Linear Trends

Figure 3 shows estimated linear trends separated by direction of travel and including round-trips as a third direction. Significant caution is urged in interpreting the meaning underlying these trends: while they could arise from physical drivers such as climate change, the commercial nature of the datasets means they could also arise due to (e.g.) adjustments to improve fuel economy and manage congestion, legislative changes around compensation due for delays relative to schedule (e.g. EC261 regulations), or more advanced operational weather modelling to e.g. avoid predicted turbulence patches.

Data after August 2023 have been removed to ensure that computed trends start and end on the same calendar month. Linear trends have then been computed from monthly median data for each season individually and for the full dataset. The magnitude of the linear trend, in units of minutes of delay per decade, is overlaid in text at the bottom of each panel.

In both the westward and round-trip directions we see a positive trend, ranging from 1.0 minutes per decade for westward flights in JJA up to 5.6 minutes per decade for westward flights in SON. Eastward trends are more mixed, with positive trends of similar magnitude to those for the westward and round trip time series in JJA and SON, but zero trend in MAM and a moderate negative trend in DJF. The full-year trend is small and positive.

### 4.2 Comparison of Extrema

We next consider measured delays for flights made when our indices are at extreme maximal or minimal values. This acts as a plausibility check on whether the indices are associated with measurable flight-time signatures at all before we move to more mathematically-derived analyses presented below. Figure 4 shows the results of this assessment.




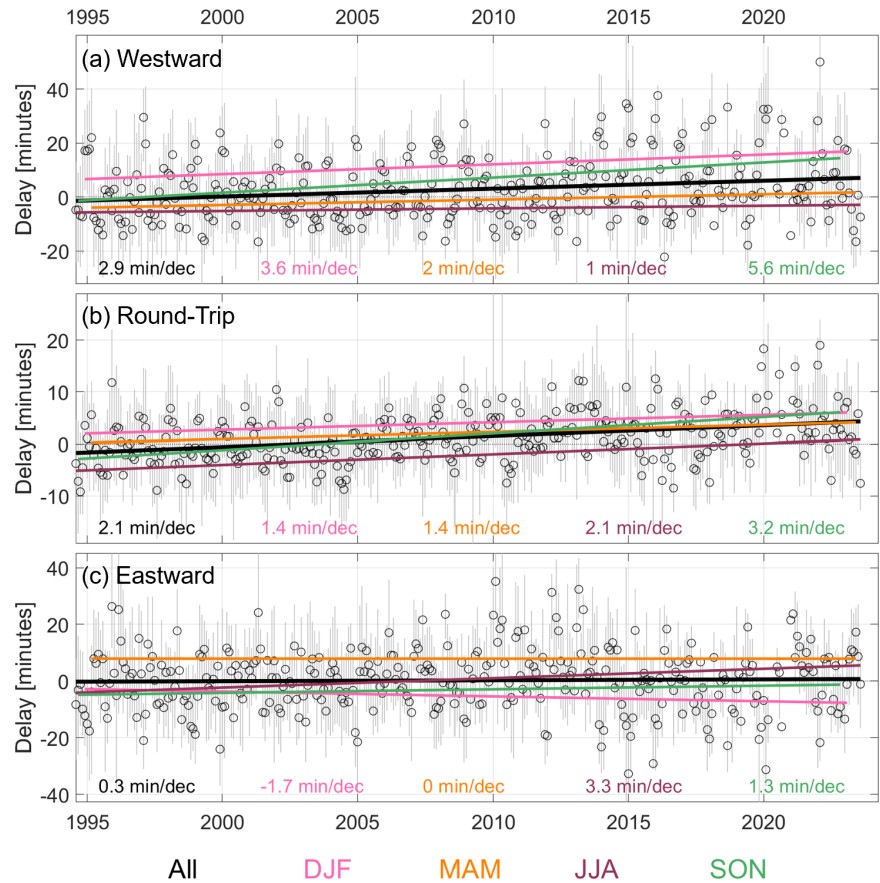

**Figure 3.** Linear trend analysis for flights in the (a) westward (b) round-trip and (c) eastward directions. Black circles indicate monthly-median delay for that month, with thin vertical grey lines showing the range between the 18th and 82nd percentiles of the data, equivalent to one standard deviation for normally-distributed data. Overlaid lines show linear trends in monthly medians for each season and for the full dataset, with the full dataset trend shown in black, DJF in pink, MAM in orange, JJA in brown and SON in green. The magnitude of the linear trend, in minutes of delay per decade, is overlaid at the bottom of each panel, with both colour and order corresponding to the key at the bottom of the figure.

We first identify all flights in each season and flight direction, then associate them with daily index values. For each index we then identify flights associated with the 5% highest and the 5% lowest values of each index, in all cases applying the lags identified in Section A1. We have tested this 5% cutoff value over the range 1–33%, which results in numerical changes consistent with our discussion below but no major structural changes in the relative form or directionality of our results.

    We next use a kernel density estimation (KDE) approach to assess our results, by computing a kernel density function (KDF)
for each distribution on a time delay scale from -60 – +60 minutes at a spacing of one minute using a Gaussian kernel. Almost identical results can be produced using Epachnikov or triangular kernels, while a simple probability distribution function (PDF)







**Figure 4.** Kernel density functions (KDFs) for the top (red) and bottom (blue) 5% of flights by each index, with panels (c,k,m) annotated to aid interpretation. KDFs are shown for flights in the (a-f) westwards and (m-r) eastwards directions and for (g-l) composite round trips. Columns from left to right show results split on the basis of (a,g,m) Annual, (b,h,n) ENSO, (c,i,o) NAO, (d,j,p) QBO, (e,k,q) Time and (f,l,r) TSI indices. Overlap between the two KDFs are shown in dark grey if the difference between the two KDFs is statistically significant at the 5% level on a two-sample Kolmogorov-Smirnoff test and light grey otherwise, with the numeric K-S test result shown at top right marked as 'PK'. The results of a two-sampled *t*-test are also shown, marked as 'PT'. Triangle markers on the horizontal axis indicate the median value in each subset of the data, using the same red/blue colour coding.





applied to the raw data gives the same results but requires wider time bins to reduce noise. For each index in each direction, we show the high-index KDF in red, and the low-index KDF in blue.

Finally, to determine statistical significance we carry out a two-sample Kolmogorov-Smirnoff (KS) test on the underlying
data, shown at the top right of each panel marked 'PK'. Values <0.001 are numerically truncated, as results for very different distributions can be numerically very small. Panels which are statistically significantly different at the 5% level have the overlap between the blue and red kernels shaded in dark grey, and those not significant at this level in light grey. A KS test is chosen to avoid the assumption that the data are normally distributed; we also show $t$-test results computed on the same data marked as 'PT', and in no case do the two tests disagree on whether a significance level falls below 5%.

We first consider the annual cycle and the linear time trend. For the annual cycle (Figures 4a,g,m), the sinusoid used maximises on the 1st of January and minimises on the 1st of July (Figure 1), and hence this analysis compares midwinter flight times to midsummer ones. We see large differences, with westward flights (Figure 4a, blue KDF) taking 17 minutes more and eastward flights 15 minutes less time (Figure 4a, red KDF) at midsummer than midwinter, both with a significant spread. Midwinter round-trips average 5 minutes longer than at midsummer (Figure 4g), but with a narrower distribution than for either
individual direction.

As discussed in Section 4.1, the full dataset exhibits a small linear trend. Our results here (Figures 4e,k,q) are consistent with that, with a difference of 5 minutes in median round-trip delay between August 1994 and March 2024, agreeing well with our estimated trend of 2 minutes/decade from the monthly-median data used in Section 4.1 when allowing for seasonal differences and the use of individual-flight data here. These differences are statistically significant in all three directional cases for both
variables.

Considering next our climate system indices, we see large differences between extreme NAO+ and NAO- conditions in both directions (Figures 4c,o), and smaller differences between strong El Niño and strong La Niña conditions (Figures 4b,n). Round trips take slightly longer at maximum NAO+ than at minimum NAO- and slightly longer at minimum La Niña than at maximum El Niño (Figures 4h,i), but the differences are small. The small effect sizes here are to some degree a product of compositing
all seasons together for this analysis, and later analyses below split by season (Section 5) show larger effect sizes for ENSO in particular.

Finally, we consider the QBO and TSI, Figures 4d,f,j,l,p,r. We see statistically-significant differences for eastward and westward trips, but not for round trips. The differences in median delay between strong QBO-westerly and strong QBO-easterly are comparable to those seen for ENSO and the annual cycle for flights in the eastward direction, at around nine
minutes, but the westward difference is smaller. Since flights typically travel near the jet centre when eastbound but not when westbound (Figure 2m-p), this may suggest a relationship between the QBO and the speed of the jet-centre winds, but an absence of such a relationship for wind speeds away from the jet.





## 5 Regression Analysis of Flight Delays

### 5.1 Multilinear Regression

To characterise the relative impacts of the processes associated with our indices on individual flights, Figure 5 shows the results of a multilinear regression analysis, subdivided by direction and season.

Delay estimates have in each case been regressed against the -1 – +1 normalised indices shown in Figure 1b-g and then doubled; this means that the results represent the change in flight time between the maximal and minimal value of the index, e.g. between maximal NAO+ and minimal NAO-. The choice to use the full range was made due to the asymmetry of some 265 indices used, particularly TSI where it is not meaningful to define a central value in time (see Figure 1g).

Since we use library code to compute the regressions it is non-trivial to apply a KS test. As such, error bars represent the standard error on the value; this is a parametric test which assumes normality, but since our results in Section 4.2 using the KS test were consistent with results from a parametric $t$-test we do not expect this to affect our conclusions.

While we do not expect perfect quantitative agreement between the results derived here and those presented in Section 4.2 270 above due to (a) the inclusion of $10\times$ as much data spanning the full range of each index and (b) the cross-linked nature of the analysis across indices, the results we see using this approach agree with those seen above in terms of direction and statistical significance with two exceptions, specifically that the delays associations between (i) westwards flights time and TSI and (ii) eastward flight times and the QBO are no longer statistically significant

We first consider time. Consistent with Sections 4.1 and 4.2, for all seasons and directions we see increasing flight times[3]; 275 in the full dataset, this averages around 8.3±0.8, 2.4±0.8 and 5.8±0.5 minutes for the westward, eastward and round-trip directions respectively over the data record.

We next consider the NAO. Consistent with Section 4.2, this has by far the largest effect. For the all-dataset case, peak NAO+ is associated with a 43.9±1.5 minute increase in westward and a 41.9±1.4 minute decrease in eastward flight times relative to minimum NAO-, netting out to a round-trip delay of 4.0±0.8 minutes[4]. In all three cases, the uncertainty is small relative 280 to signal size and the results are statistically significant. There is also large seasonal variability in NAO-associated changes, with absolute one-way effects maximising at an 82.2±3.5 minute reduction in eastward DJF flight times and minimising at a 21.3±2.2 minute increase in westward flight times in JJA. In seasonal subsets of the data however, the round trip effect is less than 5 minutes and not statistically significant. This indicates that the large changes in flight times associated with the NAO are strongly directional and have only a small residual effect on round-trips. Since our optimal lag for the NAO was found to be at 285 or near zero in all cases, this effect is effectively instantaneous.

The next largest all-dataset effect is seen for ENSO, for which we estimate that peak El Niño is associated with westward flights 11.8±0.9 minutes longer and eastward flights 7.4±0.9 minutes slower relative to minimum La Niña, allowing for the lags described above. Autumn and winter effects of ENSO are small and to within standard error bounds consistent with zero

---

[3]With one exception, JJA eastwards, which is not inconsistent with zero to within error bars, but is also not statistically significant

[4]The small deviation between the round-trip estimate and the simple difference between eastward and westward trips here is (a) within error bars and (b) not inconsistent with the data analysis approach, since not all one-way flights will contribute to a composite round-trip





**Figure 5.** Delay regression coefficients computed over all flights in (a-c) our dataset, (d-f) DJF, (g-i) MAM, (j-l) JJA and (m-p) SON, for flights in the (a,d,g,j,m) westward, (b,e,h,k,n) round-trip and (c,f,i,l,p) eastward direction. Within each panel, rows shows the regression coefficient estimated over all flights against the climate index marked at the end of the row, with bars to either side of the symbol indicating the uncertainty on that coefficient. Coloured markers with bold outlines indicate estimates significant at the 5% level, and white markers with narrower outlines non-significant estimates. Coefficients are given as delays in minutes on a typical flight duration, with negative values indicating earlier-than-average arrival. The adjusted $R^2$ of the fit combining all indices is indicated at the bottom right of each panel as text. Horizontal axis ranges have been selected to optimise for visibility of all values, and accordingly some estimates for the NAO fall off the edge of their panel; these are indicated by an arrow and numeric indicator at the end of the relevant row showing the central value, but with the marker and error bars shown as if the estimate was centred at the edge of the panel.

in all three directional cases, perhaps with the exception of autumn round-trips, and the largest effects associated with ENSO
are instead seen in spring and summer.





A positive QBO index (i.e. strong QBO-westerly) is associated with small but significant flight time increases in the westward directions at the all-dataset level and in SON, for round-trips in all seasons except SON, and in the eastwards direction in DJF. It is also associated with flight-time decreases in the eastward direction in SON, explaining the zero association with round-trips in this season.

Finally, TSI effects are associated with only small and non-significant changes for westward and round-trip flights when considering the full dataset, but solar maximum is associated with a reduction in eastward flight times relative to solar minimum of 2.7±1.1 minutes. At the seasonal level the solar cycle is also associated with flight times reductions in the westwards direction of 7.5±2.3 minutes in DJF and 7.6±2.7 minute in MAM, and increased flight times in the eastward direction in MAM of 6.3±2.5 minutes, with all other associations falling below our chosen threshold of statistical significance, i.e. 0.05 on a $t$-test.

In all cases, the adjusted $R^2$ estimator suggests that our indices describe nearly a third of the total variance, maximising at 0.27 for eastward flights in DJF primarily due to NAO+ being associated with >80 minute longer flights relative to NAO-. This initially seems low relative to a perfect prediction value of 1, but given the large number of other processes operating on flight durations is arguable quite large and this difference is consistent with both the nature of the climate system and with our measurements being made by active piloted platforms rather than passive atmospheric tracers. Climate-dynamical processes at distant locations inherently have only indirect effects on wind speeds, while routing decisions by pilots and air traffic control can have large direct effects that are both dependent on inherently limited forecast estimates of mean-flow wind speeds far away from land and will also will be made for a diverse range of other reasons.

## 6  Fuel and CO$_2$ Cost Implications

We next estimate the fuel cost and CO$_2$ emission implications of our index-associated delay estimates relative to index-median flights. For clarity of prose, we will refer to both CO$_2$ and fuel-price implications as 'costs' throughout.

We first estimate the fuel-use characteristics of each specific IAGOS aircraft (Appendix D1) and scale them up to represent US-Europe aviation as a whole using historical flight volumes (Appendix D2). A large assumption here is that these aircraft are typical for transatlantic flights; in practice, our dataset consists entirely of Airbus A340 and A330 aircraft in the range of 230 – 350 seats versus an average for all flights included in our Appendix D2 calculations of 227 seats, so our final values may slightly overestimate the true values for both fuel use and CO$_2$ production. However, the magnitude of any under-or overestimate cannot be properly quantified without more information on the relative fuel burn characteristics of the other aircraft included in this scaling factor.

To convert these values to total cost estimates, we first associate each flight with the index values for the day it took place, then use the regression-derived estimates of effect size from Section 5 to scale from the full range presented there to a flight-specific index-associated delay for each flight and, by averaging between the two legs, each round-trip. We then scale this flight-specific delay by the fuel use of the specific aircraft which made that flight, and then by the total number of transatlantic flights in that direction using passenger aircraft with >150 seats on routes of ≥4 000 km (Appendix D2). Finally, we convert





this to cost estimates in $CO_2$ and in US dollars based on this fuel use. We also compute a summed estimate of the total

325 associated with all indices for each flight.

In the below discussion (i.e. Sections 6.1 – 9) we use constant May 2023 flight volumes and fuel prices, allowing us to focus on climate process effects. May is chosen as the approximate midpoint in annual flight volumes, and 2023 as the last May in our dataset. For this specific month, these values were (a) a total of 199 flights meeting the above criteria each way per day, and (b) a price of USD 2.21 per gallon of fuel. For historical context on real costs incurred since 1994, Supplementary Figures S1

and S2 reproduce Figure 7 at real prevailing prices and monthly-mean flight volumes for $CO_2$ and financial costs respectively, but we do not discuss these figures further. Climate index values are always taken from the real calendar day in both the main and supplementary figures.

For brevity of analysis, we use our all-flights regression estimates, i.e. those in Figure 5a-c, and do not consider individual seasonal estimates. Note that, as discussed in Section 2.2, our climate indices are normalised to a range of -1 to 1 rather than to a

335 mean of zero. As a result of this choice, our cost estimates are not zero-centred, since the mean of a given index is usually offset slightly from the midpoint of the range. Since our conversion from regression-estimated delays per flight to a total monthly cost involves several significant assumptions, we also do not propagate our uncertainty analysis through this section to avoid implying certainty in these bounds.

### 6.1 Cost Distributions

Figure 6 shows the results of this analysis against the cost in $CO_2$ (top axis) and in monetary terms (bottom axis) at constant May 2023 flight volumes and fuel prices. Due to the larger data volume here compared to Section 4.2, we use simple PDFs rather than KDFs, defined using 100 bins evenly distributed across the full range of each dataset and with no smoothing. Since the relative spread of costs per plane is narrow, our estimates in this section near-directly scale as a simple linear combination of the regression estimates from Section 5 and the time-distribution of the index cycles visualised in Figure 1, but presenting

them in this combined form allows for a direct visual interpretation of the temporal spread of cost implications.

We consider first the NAO (Figures 6c,h,m). We see a well-centred PDF with slight positive skew for westward/round-trip flights and negative skew for eastward flights. The distribution is broad: for one-way flights, the 2.5th and 97.5th percentiles have a range $>100\,kT$ of $CO_2$ or 20 million US dollars per month. The more central 18-82% (1 standard deviation) spread of estimates is associated with changes in monthly flight costs of up to $40\,kT$ $CO_2$ or 10 million USD. The overall distribution

when summed over all indices (Figures 6a,f,k) is very similar in form and magnitude to that for the dominant NAO index, consistent with this being by far the largest individual cost driver.

We next consider ENSO, Figures 6b,g,l. Unlike the NAO, the distribution is quite spiky, with notable secondary maxima at large positive and negative costs. A skew is also present in both directional subsets, with the distribution extending to much larger additional cost extrema for westward flights and larger cost reduction extrema for eastward flights, and local minima are

355 present near the centre of the distribution.

Results associated with the QBO and TSI must be interpreted more cautiously, since the regression estimates used to compute them were not statistically significant for round-trips and only reached the chosen significance threshold for the QBO-westward



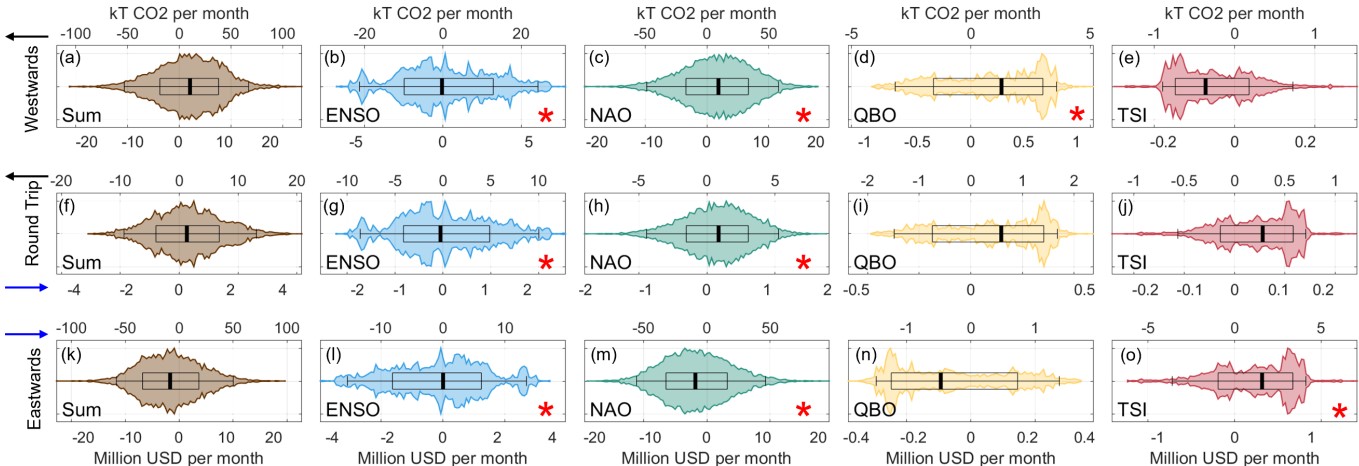

**Figure 6.** Estimated costs of climate-index-associated delays in (top axes) kilotonnes of $CO_2$ (bottom axes) millions of US dollars, using constant May 2023 flight volumes and fuel prices. Data are computed using the round-trip regression estimates shown in Figure 5, and are shown for (a-e) westward flights (f-j) round-trip flights and (k-o) eastward flights, computed separately for delays associated (b,g,l) ENSO, (c,h,m) the NAO, (d,i,n) the QBO, (e,j,o) TSI and (a,f,k) the instantaneous sum of these for each flight. Each panel shows the relative distribution of costs for each individual flight (coloured violins), as well as the distribution median (thick black line, by definition zero in all cases), 18th-82nd percentile range (box, equivalent to the one standard deviation range for normally-distributed data), and 2.5th-97.5th percentile range (whiskers, equivalent to two standard deviations). Red stars indicate that the regressors used were found to be statistically significant in Section 5.

TSI-eastward associations. Bearing this in mind, for the QBO and round-trip pair we see a very strong skew, with the spread going to very large negative values (i.e. reduced flight costs) and to only relatively small increased costs. Individual directional estimates are consistent with this in both directions. TSI estimates show similar skew for similar reasons. In all QBO and TSI cases, peak costs are small due to the very small regression coefficients estimated for the effects associated with these indices in Section 5.

## 6.2 Cost Patterns

While the distribution of each index over a long time-average is inherently zero-centred, in combination they can differ very significantly from the median for extended periods while also self-cancelling over relatively short periods. This means that meaingful long-term patterns can be seen at the bulk level. To quantify these bulk variations, Figure 7 shows monthly estimated $CO_2$ and fuel costs at May 2023 values for (a) westward (b) round-trip (c) eastward journeys over the study period. As mentioned above, versions of this figure at real-date flight volumes and fuel prices are shown in Supplementary Figures S1 and S2.

These estimates have been produced by first computing individual delay-cost estimates for each flight, then averaging to produce a monthly mean. This two-step process should reduce any effect of flights clumping within a given month, while still





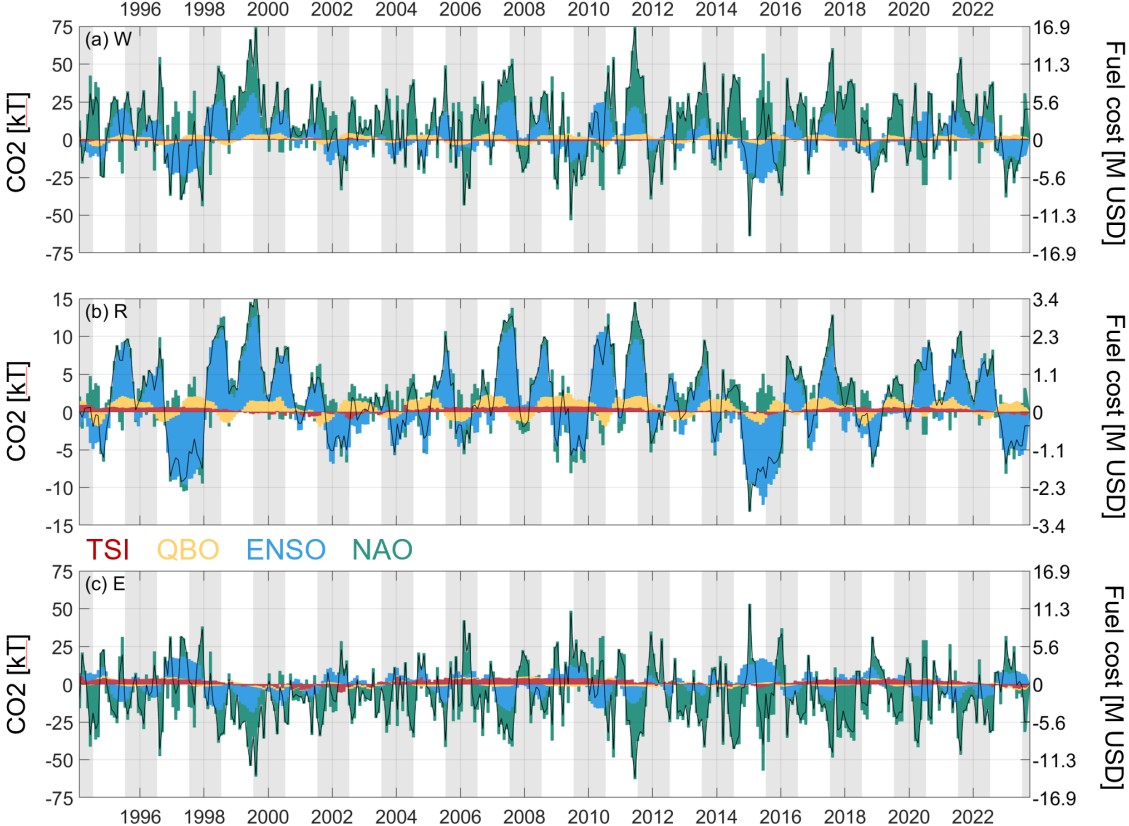

**Figure 7.** Estimated monthly costs due to climate-index-associated delays from June 1994 to March 2024, in (left axis) kilotonnes of $CO_2$ and (right axis) millions of US dollars for (a) westwards (b) round-trip (c) eastwards flights at fixed May 2023 costs and flight volumes. Data are shown as stacked histograms, i.e. the total effect due to a specific index is the difference between that and the next index in the direction of the zero axis. Indices have been ordered for this process such that the slowest-varying (TSI) is closest to the axis and the fastest-varying (NAO) is furthest from the axis. A thin black line shows the net cost summed over all indices for each month.

implicitly weighting our analysis to the actual mix of aircraft used. Each figure is shown as a stacked histogram to the total cost over all indices, with indices stacked in order of the speed at which they vary moving away from the central axis and with a black line overlaid to show net monthly costs over all four indices.

### 6.2.1 Westward and Eastward Flights

As seen above, westward and eastward estimates (Figures 7a,c) are dominated by the fast-varying NAO, the rapid cycle of which lead to frequent switches between it being associated with positive and negative costs. Peak values are lower than those seen in Section 6.1 due to the use of monthly-mean values which smooth out the rapidly-varying NAO signal.





The second largest one-directional contributions are associated with ENSO, which varies more slowly and thus in many
periods (e.g. 1999, 2007, 2015) is associated with cost changes comparable to the NAO in magnitude. QBO- and TSI-associated
contributions are much smaller, but provide a slowly-varying baseline which is potentially of practical use as it possesses
significant long-timescale predicability, and in particular TSI plays a small but meaningful role in eastward flight costs.

### 6.2.2 Round Trips

For round-trips, the extremely large variations in the NAO largely cancel out, although it can still play a large role in specific
periods, for example during 2007, 2011, 2015 and 2017. Due to this cancellation, the dominant factor in round-trip costs is
instead ENSO. While the effect size associated with ENSO for any specific flight is roughly equivalent to the NAO (Figure 5b),
the slower temporal cycle of ENSO means that its effects do not cancel at the monthly level, and accordingly most of the large
long-term changes we see in costs over are associated with ENSO, both increases (e.g. 1998-2000, 2020-2022) and decreases
(e.g. 1997-1998, 2015-2016).

Similarly, the QBO plays a much larger role for round trips. It is regularly associated with as much as 10-20% of the change
in monthly costs, and in cases where the NAO and ENSO are both weak (e.g. 2023) can act with a similar magnitude as the
usually-dominant ENSO-associated signal. It is important to note though that our estimate of delay for the QBO - round trip
association was not found to be statistically significant (Figure 5b), and this must be interpreted accordingly. Finally, TSI (again
caveated) is associated with only a negligible role in monthly round-trip costs.

These results have important implications for practical management of flight costs at seasonal to multi-year timescales,
which we discuss in Section 9.

## 7 Regression Analysis of Flight Altitudes

Since we have full traces for each flight, we are also able to quantify the association between our indices and the altitude
levels flown. This has implications for fuel use and engine design (the lower atmospheric densities at high altitude change fuel
burn characteristics) and for other climate-related effects of aviation such as contrail formation (higher altitudes are on average
drier).

To determine this, we first take each flight trace, bin it onto a 200 m vertical grid, then find the modal height for each flight;
specifically, we use the IAGOS-recorded barometric altitude for this calculation. We then repeat the regression analysis from
Section 5 but for modal altitude. This assumes that modal altitude characterises the flight sufficiently well; this is tested in
Appendix C3, concluding that it is adequate in most cases (half of flights spend at least half their time at modal altitude), but is
a weak assumption for a meaningful fraction of the data (5% of flights spend less than a third of the time at modal altitude).

The results of this are shown in Figure 8. We see altitude changes of at most hundreds of metres, with a maximal value
of 540±70 m for TSI in SON for eastward flights, Figure 8p. This initially seems small relative to typical flight altitudes of
∼10-12 km, but is fractionally large relative to the range of heights actually used, since the standard deviation of the set of
410 modal flight altitudes we use is itself only 560 m.







**Figure 8.** Modal height regression coefficients computed over all flights in (a-c) our dataset, (d-f) DJF, (g-i) MAM, (j-l) JJA and (m-p) SON, for flights in the (a,d,g,j,m) westward, (b,e,h,k,n) round-trip and (c,f,i,l,p) eastward direction. For each panel, rows shows the regression coefficient estimated over all flights against the climate index marked at the end of the row, with bars to either side of the symbol indicating the uncertainty on that coefficient. Coloured markers with bold outlines indicate estimates significant at the 5% level, and white markers with narrower outlines non-significant estimates. Coefficients are given as differences in altitude, with negative values indicating lower altitudes. The adjusted $R^2$ of the fit combining all indices is indicated at the bottom right of each panel as text.

At the all-dataset round-trip level, we see a near-zero non-significant relationship between ENSO, the NAO, the QBO and modal altitude, but a large positive association for TSI and time.

The association between TSI and flight altitude is in all cases clear - in all seasons and directions except for westward flights in DJF, solar maximum is statistically significantly associated with an increase in flight altitudes of between 130–540 m relative to solar minimum, with a small standard error (minimum 30 m, maximum 80 m) in all cases. Similarly, the relationship with





time is generally significant and positive in all cases except westwards in DJF and SON, with flights flying 130–540 m higher in 2024 than in 1994.

Considering other indices, the clearest associations are seen in JJA, when all associations except that between ENSO and eastward flight times are statistically significant and all associations are positive to within error bars, i.e. El Niño, NAO+ and QBO-westerly are all associated with higher-altitude flights than their inverses in summer. Other than in JJA, we see a statistically significant decrease in modal altitude flown at NAO+ of 410±100 m – 310±90 m relative to NAO- in all three seasons and an increase in DJF eastward flights of 260±90 m. For positive ENSO we see an increase in height in seasons for westward flights except SON, which is non-significant, and in the eastwards direction a reduciton in height in DJF and non-significant results in other seasons. Finally, for the QBO, we see an increase in height for all directions except MAM (the latter not significant) during QBO-westerly, JJA/SON round-trips and JJA westward trips, and a reduction for DJF westward trips, with all other seasons and directions non-significant.

A possible confounding effect here could be that aircraft may fly at heights relative to the tropopause (either explicitly or implicitly due to NWP), with the observed relationships thus arising from an association between tropopause height and the climate indices considered. Appendix C4 tests this, concluding that tropopause height is uncorrelated with flight altitude in our dataset and thus that this is unlikely to be the underlying mechanism.

An equivalent analysis (omitted for brevity) using 20 hPa-spaced pressure level bins instead of the height bins described above shows near-identical results. The results presented in this section are thus not significantly affected by the choice to use height as a vertical coordinate rather than pressure.

## 8 Discussion

### 8.1 Previous Work

In general interactions of this type have not been widely studied, but some reports do exist. The two most relevant studies are those of Kim et al. (2020) and Karnauskas et al. (2015). Kim et al. (2020) used ERA-Interim winds from peak NAO+ and El Niño years to simulate 810 wind-optimal London-New York and, separately, mainland US-Hawai'i (hereafter 'East Pacific') routes under extreme NAO and ENSO conditions respectively; Karnauskas et al. (2015), meanwhile, used total (i.e. takeoff to landing) flight times obtained from a dataset of 250,000 flights over the East Pacific corridor between 1995 and 2013

In the transatlantic corridor, our results agree with Kim et al. (2020) closely: specifically, they found that NAO+ increased transatlantic round-trip flight time by 4.2–9.4 minutes relative to NAO-, consistent within uncertainty range with our estimate of 4.0±0.8 minutes. Given the very different nature of the two studies, this is encouraging.

To compare results for the East Pacific corridor, we have also analysed the IAGOS dataset for flights between Honolulu and all airports in the latitude range 30°N – 70°N and longitude range 130°E – 115°E, i.e. the US and Canadian west coast. Supplementary Figures S4a-r and S4s reproduce Figures 4 and 1a respectively for these data, with the modification that we include the top and bottom 20% of data for Supplementary Figures S4a-r due to lower flight volumes.





However, the results obtained (a round-trip decrease in travel times of ∼7 minutes at El Niño vs La Niña) disagree with both previous studies. Specifically, Kim et al. (2020) estimated an increase of 5-9 – 8.7 minutes, while Karnauskas et al. (2015) did not estimate a direct value in the same terms but instead demonstrated that round-trip times along this route increased by $0.57\pm0.11$ minutes per $ms^{-1}$ of wind speed change at the 300 hPa level, with the 300 hPa winds in turn positively correlated with ENSO, i.e. an agreement of sign with Kim et al. (2020).

A likely reason for this discrepancy is the much lower volume of data on this route relative to the transatlantic route in the IAGOS dataset. After quality filtering, this corridor contains only 432 round-trip flights, all from the same individual plane and centred in three temporal windows between late 2017 and early 2024 (Supplementary Figures S4s). This is both a very small number for our analysis, and is also predominantly distributed within ENSO+ periods (Figure 1c). This strongly suggests that we cannot fully trust this result. Consistent with this, a linear regression analysis (not shown) produces results which are both not significantly significant and not inconsistent with the results presented by (Kim et al., 2020) to within error bars.

## 8.2 Limitations of our Analysis

Our results inherently have several limitations. The most important of these are (1) the lagging schema chosen, (2) the large fraction of uncontrolled variability, (3) uneven data coverage, (4) the assumption of index independence, and (5) total data volume.

1. The applied lagging is determined by a cross-correlation analysis at the full-dataset level. These choice was made to constrain the variable space, but could affect our results. We investigate this in Appendix A3, concluding that this choice, while important, is not a critically-important factor in our analysis.

2. Due to the many operational factors involved in commercial aviation, these four climate processes describe only a fraction of total variability. We quantify this in Section 2.2, but due to the imbalance between explained effect size and the full variance space it is likely that (a) some errors leak through to our results and (b) the quantification itself likely contains some error.

3. On average our dataset contains slightly over 1.5 flights per day for the period studied, but this is not even in time (Figure 1a), and 50% of the flights included are before February 2004, i.e. half of the data describes one third of the time. This is partly due to a very large pandemic-driven reduction in flights in 2020-21 when only two of the 13 aircraft in the dataset continued to fly, and coverage does return from 2022 onwards. This should not significantly affect our results for the annual cycle, NAO, ENSO and time due to the many cycles the early period covers, but could in particular impact our results for slow-varying TSI.

4. Our regression analysis assumes that the flight time response to each index of the winds controlling flight times is linear and independent, i.e. the indices do not interact. While we quantified this effect in Appendix A2, many studies (e.g Salby and Callaghan, 2000; Hansen et al., 2016; Scaife et al., 2024) have shown that the climate processes these indices describe can and do project on each other over both long and short timescales via various teleconnecting mechanisms.





5. Finally, the data volume used is lower than in some other studies. However, the nature of our dataset gives us counter-vailing benefits. Karnauskas et al. (2015) used 250 000 flight durations, but as the dataset only consisted of arrival and departure times were unable to apply flight-level quality-control filters and sensitivity tests. Tenenbaum et al. (2022), meanwhile, used $3.2 \times 10^9$ data points (compared to a total of $\sim 0.1 \times 10^9$ in this study) across an unspecified number of flights to assess jet speed changes in the Atlantic, but over a shorter period (2002-2020) and along a single route (JFK-LHR).

## 9 Conclusions and Implications

### 9.1 Conclusions

In this study, we have used 16 327 real IAGOS flight traces collected over the period 1994-2024 to quantify the change in transatlantic flight times associated with several key climate processes, specifically the El Niño-Southern Oscillation (ENSO), the North Atlantic Oscillation (NAO), the Quasi-Biennial Oscillation (QBO), the 11-year Solar cycle (TSI) and a linear time trend.

We conclude that:

1. ENSO and the NAO are associated with strong and significant changes in flight times. At the full-dataset level, peak NAO+ is associated with an 43.9±1.5 minute decrease in eastward flight times and 41.7±1.4 minute increase in westward flight times relative to minimum NAO-, increasing when considering winter only to an 82.2±3.5 minute decrease eastward and 69.9±3.6 minute increase westward. At the full-dataset level maximum El Niño is associated with an 11.8±0.9 minute decrease in westward flight times and a 7.4±0.9 minute increase in eastward flight times relative to minimum La Niña.

2. TSI and the QBO are associated with smaller but still meaningful and significant effects. High TSI is associated with decreased flight times for westward flights in winter and spring and increased flight times for eastward flights in spring, while strong QBO-westerly is associated with slower eastward flights in winter and autumn and with faster westward flights in autumn. All of these values are in the order of 5-10 minutes range across the full cycle of the climate process.

3. Strong residual effects are seen for round-trips even despite significant cancellation. At the full-dataset level, maximum El Niño is significantly associated with a 4.8±0.5 minute decrease in round-trip flight times relative to minimum La Niña and maximum NAO+ with a 4.0±0.8 increase relative to minimum NAO-

4. Depending on season and flight direction, these four climate indices plus a linear time trend since 1994 can describe up a third of the observed flight time variability. Explanatory power is lowest for round-trip flight times (≤5%) and highest for flights in winter (21% westward, 27% eastward) and spring (15% westward, 16% eastward). The majority of this signal is driven by the NAO.





5. Flight times have been getting consistently longer for westward and round-trip flights, by between 1 and 5.6 minutes per decade depending on season with the largest increases measured in autumn. This is also true for summer and autumn flights in the eastward direction. This slowing effect could be due to physical drivers such as climate change but could also be an operational choice, and we do not distinguish between these causes.

6. Flights are higher in altitude in all seasons and directions at the end of our dataset, likely due to more modern aircraft and better flight-planning capabilities, as more efficient modern aircraft usually fly higher. However, we also see an association between higher TSI and higher flown altitudes in all seasons and directions, and between positive El Niño and higher flown altitudes in spring. These changes in flight altitude seen are typically equivalent to roughly two-thirds of a standard deviation of the full set of flight altitudes excluding ascent and descent phases.

## 9.2   Cost-Planning Implications

Our results have potentially significant implications: despite explaining $\leq 5\%$ of the total round-trip variance in measured flight times (althoguh up to 27% of one-way flight times), we show that for a single aviation corridor the combined impact of these climate processes is associated with a 27 M USD or 120 kT-$CO_2$ range in monthly one-way flight costs, and a 5 M USD or 23 kT-$CO_2$ range for round-trip flights. Since our results connect these estimates directly to simple, specific and widely-measurable climate system metrics, our results hence provide the quantitive evidence needed to facilitate effective hedging
of fuel prices (reducing costs) and/or better time planning for non-essential flights (reducing $CO_2$ production). This planning benefit arises both from the direct lag effects we have measured and from the inherent predictability of the Earth system processes they describe.

Considering first the direct impacts once the climate process state is measured, our estimates of lag-lead time (Appendix A1) suggest that the time taken for flight times to be affected can be quite large. While for the NAO effects appear to be near-instant,
we estimate that the lead time between measurement and impact for ENSO is ∼0-30 days, for the QBO is 6-9 months, and for TSI is of order a year. Thus, even absent the ability to predict these processes, their direct lead time is sufficient to allow significant planning.

In practice however, the available lead time is likely significantly longer than these estimates, due to the baseline predictability of these processes and broader scientific interest in enhancing this predictability:

– While TSI only weakly affects costs, it can be predicted to some degree a decade or more in advance.

– More usefully the QBO, which is associated with round-trip costs of several million USD or tens of millions of kT of $CO_2$ per year (Figure 7b) is a highly-regular and slowly-evolving process which can be predicted many years in advance (Scaife et al., 2014), albeit with some unusual hiccups in the last decade (Osprey et al., 2016; Anstey et al., 2021a).

– For ENSO, recent work suggests that it may be predictable at multi-year scales (Zhao et al., 2024), and due to its broad
influence on many other climate processes extending this is a major target of active research.



– Finally, the NAO is harder to predict than the above processes and also acts more quickly, but is currently predictable at timescales of 1–3 months and due to its broader impacts on northern hemisphere weather is a perennial and major research target (e.g. Collingwood et al., 2024, and references therein).

Therefore, our results provide the information needed to support flight cost planning at increasingly long lead times as our
ability to predict climate system variability continues to advance.




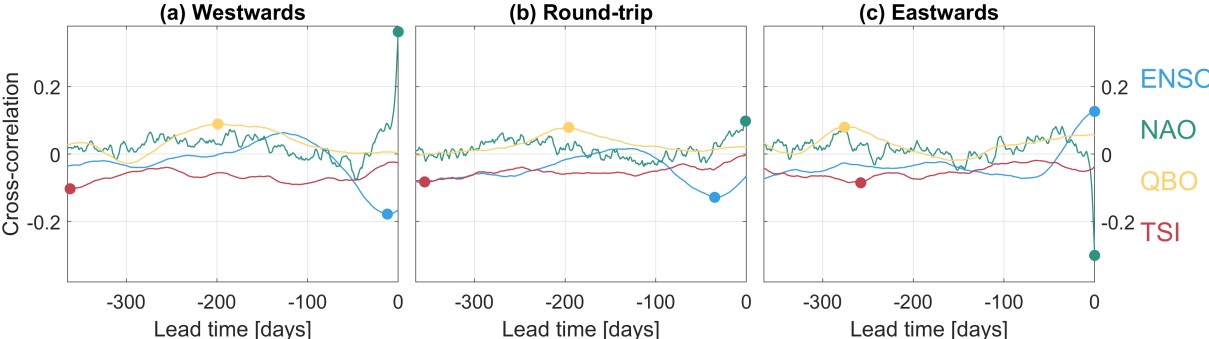

**Figure A1.** Results of a cross-correlation analysis of daily-mean flight times against four climate indices for (a) westwards (b) round-trip (c) eastwards flights. In each case, the maximum absolute value of the cross-correlation, which we take as the optimal lag for analysis for that dataset, is indicated by a circular marker.

## Appendix A: Climate Index Stability

### A1 Index Lag

To identify the appropriate lag for each index, we have carried out a cross-correlation analysis of the full flight times dataset with each index for lags of up to one year, shown in Figure A1.

Each panel shows the cross-correlation between daily values of each index and daily-median flight times in each direction of travel, with each index tested independently. For each comparison, we identify the maximum absolute cross-correlation value as the optimal lag for the dataset and use this in our analyses; the choice of absolute values is to permit strong anticorrelations. We do not test or lag the annual cycle or linear time trend.

For the NAO, the optimal lag in both directions is zero, consistent with the local process this index describes. The best-fit for

round trips peaks at one day, but this is not inconsistent since our definition of a round-trip allows for up to a day of difference between the two legs. The magnitude of the peak NAO cross-correlation is larger than for our other indices for eastwards and westwards flights, reaching absolute values >0.3 compared to maximal absolute values <0.2 for all other indices, and is positive for westward and negative for eastward flights, netting out to a small positive value for round-trips.

ENSO (blue lines) is also consistent for the three directions and also shows a fast response, with a twelve-day lead for

westward flights, 35-day lead for round-trips, and zero-day lead for eastward flights. This is perhaps an unexpectedly small lag in all three cases given the large distance to the ENSO source region, and differs for example from the relationship between gravity waves over this region in IAGOS data and ENSO studied by Wright and Banyard (2020), which peaked at 7 months. However, we do see a secondary maximum in the absolute ENSO cross-correlation results for both eastward and westward flights at around 200-250 days lead time consistent with this earlier work. Maximal values are again reversed between the

eastward and westward directions of flight.



Results for the QBO (yellow lines) suggest long lead times are involved, with maximal correlations obtained at lags of 199 days for westward flights, 196 days for round-trips, and 276 days for eastward flights. The relatively large deviation for eastward flights is consistent with previous studies.

Finally, TSI (red lines) exhibits the strongest correlation at very large lags, with best-fit values found at 362, 355 and 258 days for the westward, round-trip and eastward directions respectively. We note that tests at longer lags (omitted for brevity) suggest that this relationship may maximise at even longer periods; however, for consistency of analysis and to maximise available data when shifting the input time series, we limit our analysis to one year here.

## A2 Index Independence

In Sections 5 and 7, we use multilinear regression techniques, and accordingly must ensure that the indices are sufficiently independent. Building on our previous work in Noble et al. (2024), we do this in three ways:

1. To assess multicollinearity, we estimate the Variance Inflation Factor (VIF, e.g. Montgomery et al., 2012). Over the period studied, the VIFs of each of our indices relative to the others all lie between 1.0073 – 1.0418, where 1 is the minimum possible value, 5 a typical benchmark for concern and over 10 indicates significant multicollinearity.

2. To assess autocorrelation, we apply the statistical test described by Durbin and Watson (1950). For this test, output values are between 0 and 4, where 2 indicates no autocorrelation and values within the range 1.5-2.5 are considered acceptable. Analysis of combined eastward and westward flights gives an estimate of 2.0017, of westward-only flights 1.7271 and for eastward flights 1.7067, all well within the safe range for assuming our inputs are not lag-autocorrelated.

3. Finally, and distinctly from Noble et al. (2024), we apply the multiple collinearity diagnostics of Belsley et al. (1980) to the indices. These tests return near-dependency values between 0.0012 – 0.6412; for this test, values of $\sim$5–10 represent weak cross-dependencies and above 10 high.

Based on these tests, we conclude that these indices should be sufficiently independent for multilinear regression use.

## A3 Index Lag Effects

The lagging we apply to our indices is entirely determined via a simple cross-correlation analysis (Appendix A1) and applied uniformly at the full-dataset level. This choice was made to constrain the variable space, but could affect our results. To assess the impact of this choice, Supplementary Figure S3 shows the same analysis as that shown in Figure 5 but with zero lag applied to all datasets. We see that:

– NAO and Time results are near-identical as zero lag was applied in the main analysis. Small numerical differences are seen due to internal interactions in the regression analysis.

– ENSO results are very similar in both form and magnitude. This is reassuring given the major role of ENSO in driving our cost estimates, and is conceptually consistent with the relatively small lag used (0-35 days) combined with the relatively long timescales of ENSO changes (Figure 1c.)





– Results for the QBO are usually reversed in sign but of broadly similar magnitude. This may be an effect of the steadily-descending morphology form of the QBO and the applied lags being in the range of 22-32% of the typical QBO cycle.

– Finally, TSI results differ in magnitude to Figure 5 and are much less likely to reach statistical significance, but in most cases agree in sign.

Thus, we conclude that the effects of the lagging process, while important, do not dominate our results.

## Appendix B: The Atlantic Multidecadal Oscillation

Tenenbaum et al. (2022) suggested a role for the theorised Atlantic Multidecadal Oscillation (AMO) in transatlantic flight times, although they were unable to draw quantitative conclusions due to the end date of their data relative to AMO variability. However, recent work by Mann et al. (2021) suggests that the signature ascribed to the AMO may be an artefact of pre-industrial volcanic activity rather than a true metric of climate system variability. Accordingly, we do not include the AMO in our primary analyses.

However, to address the question raised by Tenenbaum et al. (2022), we have also repeated our regression analyses including the AMO index as described by Enfield et al. (2001). The results of this analysis, not shown, suggest a moderate effect size slightly larger than ENSO, increasing round-trip flight times by approximately six minutes, but with much stronger effects on eastward flights ($\sim$20 minutes with statistical significance) than westward ($\sim$5 minutes but without statistical significance). Total adjusted $R^2$ marginally increases for round-trips in SON (from 0.03 to 0.05) and eastward trips in JJA and SON (by 0.01 in both cases), but in all other cases the adjusted $R^2$ is unchanged to two decimal places.

## Appendix C: IAGOS Sensitivity Testing

### C1 Airport Exclusion Radius

Aircraft near an airport can take quite varied routes depending on local conditions, and this may affect our results. However, it is also plausible for this to have no effect, as the remaining flight time could be adapted to compensate for experienced and/or expected delays. To assess this effect, we carried out a sensitivity test by excluding data within a given radius of the arrival or departure airport, which we refer to as the 'airport exclusion radius' (AER).

The AER was varied systematically over the range 0–1000 km, and the adjusted data passed through our standard analysis chain to produce normalised flight time estimates. For each AER, we computed the variance of the remaining normalised flight times, taking 10 000 bootstrapped estimates of this variance as an estimate of uncertainty. Figure C1a shows the results of this analysis.

When the AER is very low ($<$1 km, i.e. the left side of the figure), we see extreme variations in estimated flight times, with both a large variance across the dataset and an extremely wide confidence interval on the variance, both indicating very noisy data. At AERs $>$50 km, meanwhile, we also see increasing variance and decreasing confidence. However, in the range between




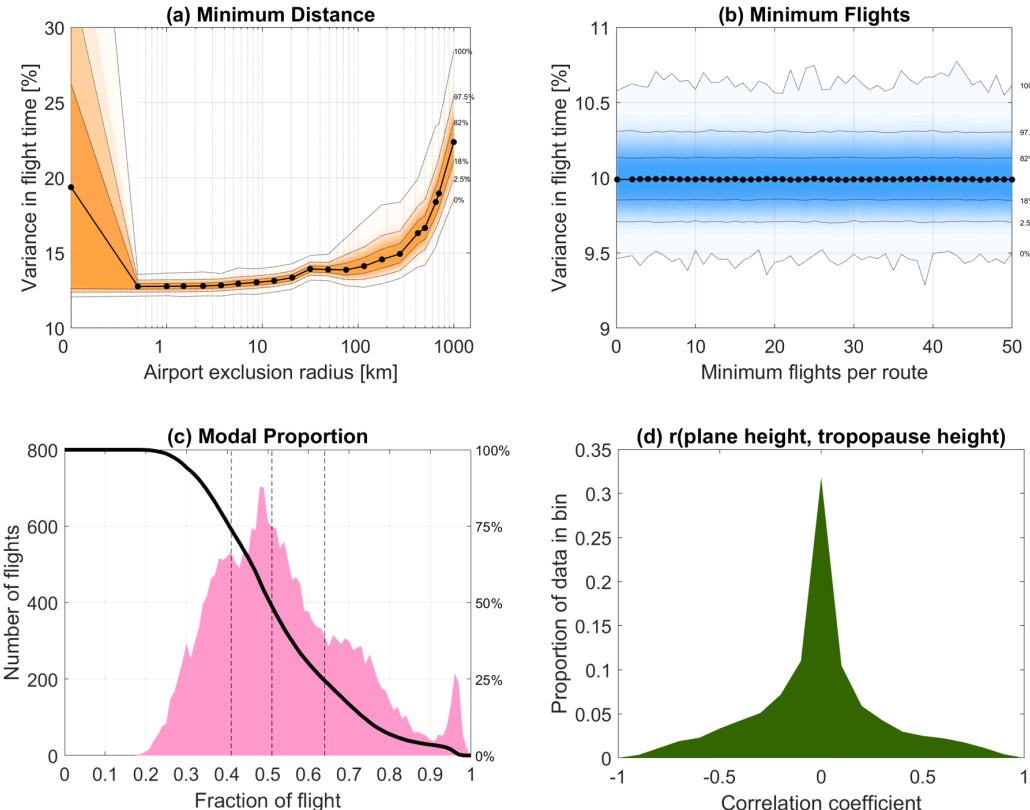

**Figure C1.** Results of the sensitivity tests for (a) airport exclusion radius, with sample radii on the horizontal axis, (b) minimum flights per route, with the minimum number of flights on the horizontal axis, (c) fractions of each flight at the modal altitude, with fraction on the horizontal axis, and (d) computed Pearson correlation between tropopause height and flight altitude, with correlation on the horizontal axis. In (a) and (b), the vertical axis shows the variance of the remaining dataset, with the central variance estimate plotted as a solid line and bootstrapped confidence intervals on this variance plotted as shading and dotted lines. Filled black dots indicate sampling points where values were computed. In (a), due to the use of a logarithmic horizontal axis, the estimated variance and confidence range for an airport exclusion radius of 0 km has been plotted at a value of 0.1 km, with gridlines removed between these value and the first true value of 0.5 km to highlight this distinction. In (c), the black curve shows the cumulative density function of the data, inverted to maximise at zero flight fraction.

0.5 – 30 km, the variance in flight time is low with high confidence, indicative of a more stable dataset. On this basis, we set our AER to 10 km.

## C2  Minimum Flights Per Route

Since our analysis normalises all flight durations to the route-median, it may be sensitive to the cutoff value chosen for the minimum number of flights required to include a given route. To take an extreme case, a route with a single flight will by definition have no delay, pulling our overall result towards the median.





To assess the impact of this choice, we have systematically varied the minimum number of flights over the range 0–50, assessing our results in the same way as the above AER testing. These results are shown in Figure C1b, and show our analyses
are almost completely insensitive to the value chosen, presumably due to the dominance of frequently-flown routes on the overall dataset. We arbitrarily choose a value of 10, primarily to restrict the set of included airports to a easily-interpretable number of cases in Supplementary Table 1 and Figure 2a,b.

## C3    Modal Altitude Validity

In our analysis of the relationship between flight altitude and climate index values, we use the modal 200 m band within which
each flight track was recorded as flying and assume this adequately characterises the altitude of the flight as a whole.

To contextualise this assumption, Figure C1c shows the proportion of time for each flight spent in the modal altitude bin. This is represented as a histogram with 1% bin size (pink shading, left axis) and as a line plot (solid black line, right axis) showing the proportion of flights which spend this fraction or more of their flight time in the modal altitude bin.

Overlaid on the line plot are three vertical dashed lines representing, in order from left to right, the 25th, 50th and 75th
percentiles of the data. From left to right, these dashed lines show that (41%, 51%, 65%) of flights spend at least (75%, 50%, 25%) of their duration outside their modal altitude bin. The central value tells us that approximately half of all flights spend half their flight time or more in their modal height bin, while the two other values tell us that a quarter of flights spend 65% or more of their time in their modal bin, but that another quarter of flights spend at most 41% of time in the modal bin.

41% seems a reasonable proportion to support the assumption that the modal altitude characterises the flight duration well,
since it still implies almost half of the flight duration being spent at a single altitude, which is likely to be enough time for the physical processes our climate indices describe to act upon the plane. However, the histogram does show some very low values, with five percent of flights spending less than a third of their duration at modal altitude. Thus, while the modal altitude assumption adequately characterises a large fraction of the data, it should be treated with some caution.

## C4    Correlations between Tropopause and Aircraft Altitude

A possible source of the apparent dependence of aircraft altitude on the climate indices we consider could be due to aircraft adjusting their height in response to tropopause height changes also driven by the climate indices. To test this, we have computed an estimate of the tropopause height at every point in every IAGOS flight using ERA5 temperature data, following the method of Reichler et al. (2003). ERA5 data are assumed to be suitable for this purpose as a separate assessment (omitted for brevity) comparing ERA5 winds to IAGOS-measured winds shows correlations >0.95 for almost all flights. This is expected,
and is consistent with the assimilation of the aircraft data into ERA5.

For every flight, we use these estimates to compute a Pearson linear correlation coefficient between tropopause height and flight altitude over the middle third of each flight. The results of this assessment are shown as a histogram in Figure C1d. The distribution of measured correlations peaks strongly at zero with only small wings at high correlations and anticorrelations, consistent with a lack of strong relationship between flight altitude and tropopause altitude in the IAGOS dataset.





## Appendix D: Cost Scaling Factors

### D1   $CO_2$ and Fuel Price Calculations

In Section 6, we calculate the extra $CO_2$ emitted and financial costs incurred due to measured delays. These calculations are carried out for the specific aircraft and flights included in our dataset, identified using aircraft tail numbers.

We first identify the model of each plane via the crowdsourced data available at Planespotter.net (2024), and using this information obtain the maximum take-off weight, maximum zero-fuel weight, typical passenger load range and maximum fuel weight for each model (Airbus, 2024).

Using these weights and the midpoint passenger load value for each model, we then use the theoretical approach described by Burzlaff (2017a) (as implemented by the spreadsheet of Burzlaff, 2017b) to estimate fuel use per kilometre at the midpoint of a transatlantic cruise, which we take to represent an average across the cruise phase of the flight. We assume a per-passenger net weight including luggage of 95 kg, an aircraft speed of 250 ms$^{-1}$, and a flight distance of 7500 km. The flight speed was estimated by calculating the average speed over the middle third of each flight (249$\pm$3 ms$^{-1}$, with the uncertainty representing a range of one standard deviation), and the flight distance as the length of the modal route travelled in our dataset, i.e. FRA to ATL. This calculation produces fuel use estimates in the range of 109 – 208 litres per flight-minute depending on the specific aircraft model.

To compute additional $CO_2$ emissions, we use a scaling factor of 3.15 kg-$CO_2$ per kg of jet fuel (European Environment Agency, 2023; US Energy Administration, 2024a). This gives estimates of additional $CO_2$ emissions in the range 281 – 537 kg of $CO_2$ per additional flight-minute depending on the specific model.

Separately, to compute fuel costs we use a monthly-mean time series of Gulf Coast Kerosene-Type Jet Fuel Spot Prices since 1991 (US Energy Administration, 2024b). In practice airlines will hedge their fuel costs, and hence this is an approximation. By calculating the speed of each flight and using the price per minute for each plane as computed above, this allows us to directly estimate the additional fuel price in US dollars for a specific aircraft on a specific day due to the effects of the climate processes our indices characterise.

### D2   Total Flight Estimates

In Section 6, we estimate the total impact of these climate indices over all flights. To do so, we use the total number of transatlantic flights as recorded by US Bureau of Transport Statistics (2024). Specifically, we use monthly data for jet aircraft (aircraft groups 6-8) which primarily carry passengers (aircraft configuration 1), have at least 150 seats on average for each flight along the route, and travel along a route of at least 4000 km.

The BTS dataset includes all international flight segments originating from or arriving into the United States, but does not include data for flights between Canada and Europe. Due to the global nature of the dataset some geographic filtering is required to be useful for our purposes, and accordingly we restrict the data to all flights with an origin/destination in the contiguous United States (World Area Codes 10-99) and a corresponding destination/origin in Europe (World Area Codes 900-999).



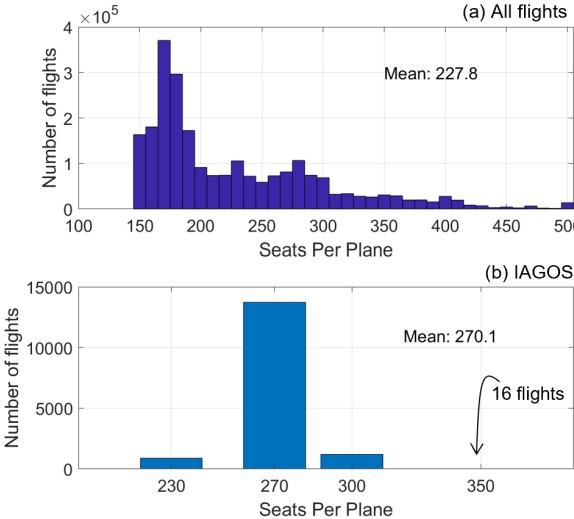

**Figure D1.** Estimated number of seats per flight by aircraft (a) with more than 150 seats per plane on each monthly-summed route in the full US Bureau of Transport Statistics (2024) dataset and (b) in our dataset.

On the North American side this is a poor match for the study region used in other sections of this study, as it includes many regions we do not study in the western United States but does not include our Canadian destinations. While these two errors will compensate to some degree, the results should nevertheless be taken as an approximation rather than an absolute value.

Finally, to assess how typical these aircraft are for this route, we have computed the number of seats per plane on each route considered in this analysis. Figure D1 shows a histogram of these results; we see a distribution skewed to relatively small aircraft, but with a mean of 227 seats per plane. This suggests that our IAGOS aircraft, with capacities of order 230-370, are larger than average.

*Code and data availability.* IAGOS data are available from the IAGOS Data Portal (Boulanger et al., 2019). All climate indices used are
705 either publicly available (NAO, TSI, ENSO) or can be computed directly from publicly-available data (QBO, Annual, Time); their source or computation method is described at first mention in the text. All analysis and plotting code are available in their current working (i.e. non-finalised) form via https://github.com/corwin365/20201122IagosTimeDifference, but if the paper is accepted will be moved to a permanent archived repository, with this text updated to link to the archived form.

*Author contributions.* CW: conceptualisation, data curation, formal analysis, investigation, methodology, software, validation, writing –
710 original draft, writing – review & editing. All others: writing - review & editing.



*Competing interests.* The authors have no competing interests.

*Acknowledgements.* CW was funded during this work by NERC grants NE/S00985X/1, NE/V01837X/1, NE/W003201/1 and NE/Z50399X/1 and by Royal Society University Research Fellowship URF/R/221023. PE was also funded by NERC grants NE/V01837X/1 and NE/W003201/1. We would also like to acknowledge the contributions of Hannah Clark, who suggested testing for the effect of latitude/tropopause fraction
and comparing results to the Hawaii-USA corridor, Scott Osprey, who suggested testing for the relationship between tropopause height and flown altitude, and Ed Gryspeerdt who provided useful comments on a draft version.

MOZAIC/CARIBIC/IAGOS data were created with support from the European Commission, national agencies in Germany (BMBF), France (MESR), and the UK (NERC), and the IAGOS member institutions (http://www.iagos.org/partners). The participating airlines (Lufthansa, Air France, Austrian, China Airlines, Hawaiian Airlines, Air Canada, Iberia, Eurowings Discover, Cathay Pacific, Air Namibia,
Sabena) supported IAGOS by carrying the measurement equipment free of charge since 1994. The data are available at http://www.iagos.fr thanks to additional support from AERIS.



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
