# Peer review of "The influence of climate variability on transatlantic flight times"

_EGUsphere, 2025_

## Author Response (AR1)

**Response to Reviewers**

First, sincere apologies for the inordinate time this response and the associated revisions have taken to produce - fieldwork and other work travel commitments of the lead author are the reason for this delayed response. We appreciate that as a result of this delay this manuscript is very much no longer fresh in the minds of the Reviewers and Editor, and are sorry for this inconvenience.

**Reviewer 1 - Kristian Strommen**

**1. Should the anomalous duration of a round-trip equal ... the sum of westward and eastward anomalies? ... If this is not the case, then can you explain why?**

As flagged in the online discussion, this arises due to the mathematics of how wind and flight directions interact, and we have added explanatory text on this topic. This is discussed further in the response to Reviewer 3 (Mark Baldwin).

**2. L301: "In all cases, the adjusted $R^2$ estimator suggests that our indices describe nearly a third of total variance" ... Your sentence here therefore seems to oversell the actual results quite strongly, especially the way you say "in all cases"! You could rather say that the values range from 3 to 27% explained variance, with an average of 10%.**

The 'in all cases' is very clearly incorrect here - this is most likely an error introduced during text-editing, and has been removed. Thank you for spotting and flagging it! The paragraph has also been modified to say "as much as a quarter", as this is indeed a much fairer description than a third, and to also mention the cases where the value is low.

**3. On page 4 where you introduce the NAO index, you should say that a positive NAO is also associated with higher jet speeds, not just latitudinal shifts.**

This is a useful addition, and has been added to the text as suggested.

**4. I was left wondering if all the indices you consider affect flight times primarily by modulating the speed of the jet. One could imagine testing this by computing Woollings' jet speed index and seeing how much of all the effects can be explained by variations in jet speed. I think this might be beyond scope, so could be left as a question for follow-up work. Some brief discussion on this point in the Discussion at the end would, in any case, be nice.**

This is an interesting suggestion. To test this, we have reproduced jet latitude and jet speed indices consistent with those of Woollings et al. 2010 but modified slightly to take advantage of locally available data - specifically, we use ERA5 rather than ERA-Interim, on model levels rather than pressure levels, at a resolution of $1.5°$ and 3 hours. As with Woollings et al. 2010, we use a ten-day Lanczos filter working in 61-day chunks - the window length is less important with modern computing capabilities as we could operate at the whole dataset level, but reduces the chance of differences being introduced between our version and the originals.

Interestingly, the relationships (shown in the table below) between these new indices and the original four are maybe not as close as might intuitively be expected. This is perhaps consistent with Woollings and Blackburn 2012, who required a combination of both the NAO and the East Atlantic pattern to still explain only 62% of the spread between jets in CMIP3 models.

Considering the relationships quantitatively, the Durbin-Watson and Belsley tests both return quite small values which are only marginally changed from those in the original manuscript.

| Test | Without jet indices | With jet indices | Typical safe range |
| --- | --- | --- | --- |
| VIF | 1.0073 - 1.0418 | 1.0183 - 4.2110 | <5 |
| Durbin-Watson | 1.7067 - 2.0017 | 1.7115 - 2.0020 | 1.5-2.5 |
| Belsley | 0.0012 - 0.6412 | $6\times10^{-7}$ - 0.9242 | <5 |

However the VIF has increased by a large amount and is now edging close to where concern is usually warranted when using this test - the typical cutoff at 5 is a rule-of-thumb rather than a binary flag and affects how much trust can be put in the results, so some additional caution is warranted here given the large fractional increase.

On this basis, it is probably a little risky to include these new results in the primary narrative, but well worth including them in the study overall. The two new indices have therefore been included in a new Appendix summarising the above information and showing a modified form of Figure 5 but including the jet indices, so the reader can draw their own conclusions about whether they are sufficiently independent for inclusion.

For clarity, including this information as an Appendix rather than in the main narrative is very much a choice and we are happy to reconsider this if the Reviewers and Editor feel strongly. We did give serious consideration to including them in the primary study in this revision, but given almost every number in the study would change by a small amount, we judged that the risk of introducing subtle-but-cumulative inconsistencies and errors was greater than the benefit added by the additional explanatory power these indices provide.

**Reviewer 2 - Anonymous**

**The main concern ... is separating the dynamical signals given by the regressions against the various climate indices, and the extent to which the aircraft operators are flexible to respond to these changes given the many operational factors... Figure 2 shows a large lateral spread in the tracks. I think that it is easy to believe that a lateral shift in the flight tracks reflects climatic variability and impacts crossing times. It might also be possible that the preferred flight level is also adapted to take account of favorable or unfavorable conditions but I wonder if the flight-levels are more operationally constrained than the lateral flight tracks. Therefore, my main concern comes at the end of the end of the article regarding the discussion on the regression of flight altitudes (section 7) as it is not clear to me how this analysis works given the step-wise climbs and the nature of the fixed flight-levels.**

**At the start of a flight when the aircraft is heavy, the flight-altitude is lower. As fuel is burned and the aircraft becomes lighter, the pilots are able to make step-wise climbs until the final maximum altitude is reached. The flight altitudes are constrained by the flight separation minima required for safety and there are well established and fixed flight-levels on which aircraft operate (eastbound -odd flight levels, westbound even flight levels). Thouret et al. (1998) (JGR, vol 103, pages 25653- 25679) give a short description of the flight-levels. The point on the trajectory at which the aircraft is able to make the step-wise climb depends on the engine performance (which changes with altitude) and load-factors as well as the air-traffic control regulations. Engine performance has evolved over time, and load-factors have changed over time being sensitive to demand ( e.g. more passengers in summer than in winter leading to an annual cycle) and changing after COVID or economic cycles.**

**I think that more analysis is necessary in section 7, to untangle these different factors (engine performance, load-factors, air traffic control requirements) and determine anything useful. My suggestion would be either to discuss more about flight opera-**

**tions in the article which would explain some of these issues to the reader, or simply remove this section on flight altitudes as currently I think that the physical link with climate indices, particulary TSI and ENSO in spring (conclusion 9.1 point 6) is rather weak i.e. the aircraft are not flying higher because of the solar cycle (TSI) or EL Nino in spring but because of something else or an artefact in the data.**

The concerns of the Reviewer (and, to a lesser degree, Reviewer 3) here are interesting and technically valid, and strongly suggest that the material on altitude needs some more thought before final publication. For some additional context, we were already concerned before submission of the initial manuscript that this material was a diversion for the reader from the core narrative of the manuscript, which is otherwise fully focused on flight times. Therefore, it seems that the easiest solution is to remove this material from the current manuscript and consider it for separate publication after more detailed investigation following the guidance set out by the Reviewer - this satisfies both the scientific detail and structural narrative issues.

**Reviewer 3 - Mark Baldwin**

**1) I think that an important addition for most readers would be to add a brief discussion explaining that the RT flight time will always increase if there is an increase in the background wind speed in either direction.** *(details of mechanism omitted here but available in original review online).* **This would help to explain a lot of your results and may help at the end of the paper to explain differences with previous publications. Of course, actual flight paths attempt to mitigate this.**

Absolutely correct - thank you for this clear explanation. An explanation of the concept has been added to Section 3.3 where the round-trip methodology has been introduced, and the concept has been integrated into our discussion of the results later in the paper.

**2) Limited data set. I looked up approximately how many Europe - US flights actually took place between August 1994 and March 2024. There were approximately 7-9 million direct transatlantic flights operated between Europe and the United States. This estimate is based on analysis of available flight data, historical trends, and consideration of major aviation disruptions during this 30-year period.he data used in this study neglects 99.8% of the actual transatlantic flights. This makes me wonder how the results might differ if all the flights were available for analysis. You do mention this in 8.2 L461, but I think that you should state how small the fraction of flights available to analyze was compared to the number that took place.**

Firstly, we concur on the relatively small proportion of total flights considered in this study. To provide more precise numbers in aid of discussion: after filtering for flight type and aircraft size as described in Appendix D2 (now E2), the US Bureau of Transport Statistics dataset we use contains 3.79 million (one-way) flights for the period January 1994 to July 2024, i.e. it is in the same order of magnitude as your quick estimate but slightly smaller. Our 16 327 flights therefore represent 0.43% of total similar flights.

Even assuming the flights are fully independent statistically, which they are likely not, this would be an important factor. Therefore, it is important to strongly caveat our results, which we now do in Section 2.1, and refer back to in several other places.

**3) The underlying question is: how does the atmosphere change with the 4 indices (plus trend) and how does this affect the optimal flight times for round trip transatlantic flights? ... I suggest a comparison to a very simple theoretical flight. The basic idea would be to select several city pairs, and simply assume that flights were at**

**some constant altitude and air speed. Using reanalysis (4X daily?) simply calculate the great circle flight path time in both directions at the reanalysis times. This comparison might reinforce the results you have obtained. I expect annual, ENSO, and NAO results to be verified. But trends would be very interesting. This approach could help sort out the Honolulu results that differ from previous publications (Discussion, L444). Stronger winds should increase the RT flight times.**

This is a really good suggestion, and we agree that an atmospheric-only benchmark is useful. However, we worry that adding material on this topic would introduce significant overlap with the work of Kim et al. 2020, who applied this approach to 28 years of wind-optimal simulated flights on the JFK-LHR and HNL-SFO routes based on ERA-Interim winds, considering only the ENSO and NAO indices. Their results were briefly mentioned in our original discussion, but as this reviewer comment identifies this was clearly understated and should be better integrated into the text. Accordingly, in the revision we have mentioned this prior work more prominently in the introductory material, hopefully addressing the Reviewer's concerns. We are very happy to reconsider this and repeat the analysis ourselves for the same time period as our current study and for our full set of climate indices if the Reviewer/Editor feel this would enhance our study further.

**4) Please convert the solar cycle altitude result to meters different (not just standard deviation) in the Abstract Line 12. On line 415 you do discuss this, but I am left wondering if this effect is real.**

In response to Reviewer 2's concerns, this material has now been removed. Nevertheless, we do agree with this comment, and if we had not removed the material would have made this change as suggested.

**5) This suggestion is probably for a future paper, since it would be fairly involved: Use reanalysis to calculate millions of idealised RT flights, assuming all flights are optimised to minimise flight times. That way you would eliminate all the operational issues and address only the atmosphere. It would focus on how the atmosphere changes for ENSO, NAO, TSI etc.**

This would indeed be a very interesting study, and builds well upon other studies in the literature (e.g. the aforementioned Kim et al. 2020, but also to some degree Boucher et al. 2023) which provide the methods needed for such optimal-route sampling. As you say, it would be fairly involved, but maybe not too unmanageable - if the synthetic flights were outputted to the same format as the IAGOS ones then the existing code could be used for the remainder of the analysis. However, a larger issue is that adding this would make the narrative much more complex. Accordingly, we concur with the Reviewer that this is better suited for future work.

**References**

Boucher, Olivier et al. (Aug. 2023). "Comparison of Actual and Time-Optimized Flight Trajectories in the Context of the In-Service Aircraft for a Global Observing System (IAGOS) Programme". In: *Aerospace* 10.9, p. 744. DOI: 10.3390/aerospace10090744.

Kim, Jung-Hoon et al. (Oct. 2020). "Impact of climate variabilities on trans-oceanic flight times and emissions during strong NAO and ENSO phases". In: *Environmental Research Letters* 15.10, p. 105017. DOI: 10.1088/1748-9326/abaa77.

Woollings, Tim and Mike Blackburn (Feb. 2012). "The North Atlantic Jet Stream under Climate Change and Its Relation to the NAO and EA Patterns". In: *Journal of Climate* 25.3, 886–902. DOI: 10.1175/jcli-d-11-00087.1.

Woollings, Tim et al. (Apr. 2010). "Variability of the North Atlantic eddy-driven jet stream: Variability of the North Atlantic Jet Stream". In: *Quarterly Journal of the Royal Meteorological Society* 136.649, 856–868. DOI: 10.1002/qj.625.

---

## Author Response (AR2)

Thank you for your positive decision. Figure A1 has been moved to below the Appendices (I think this is the issue? I've read the manuscript composition document but I'm still unsure!) and ENSO etc will be expanded out in the modified short summary after submitting this statement.

Four small additional changes have been made to the manuscript:

- two typological errors have been fixed
- a broken cross-reference from removal of the altitude dependence material has been removed
- Figure C1 has been replaced with the correct image - the previous version had a duplicate copy of main Figure 5, the error arising because they look visually similar and also have very similar filenames locally.